# Stability of gut microbiome after COVID-19 vaccination in healthy and immuno-compromised individuals

Rebecca H Boston[1] , Rui Guan[1] , Lajos Kalmar[1] , Sina Beier[1] , Emily C Horner[1] , Nonantzin Beristain-Covarrubias[1] , Juan Carlos Yam-Puc[1] , Pehuén Pereyra Gerber[2,3], Luisa Faria[1], Anna Kuroshchenkova[1], Anna E Lindell[1] , Sonja Blasche[1] , Andrea Correa-Noguera[4], Anne Elmer[5], Caroline Saunders[5], Areti Bermperi[5], Sherly Jose[5], Nathalie Kingston[6], CITIID-NIHR COVID-19 BioResource Collaboration*, Sofia Grigoriadou[7], Emily Staples[1], Matthew S Buckland[7,8] , Sara Lear[4], Nicholas J Matheson[2,3,9], Vladimir Benes[10] , Christine Parkinson[4], James ED Thaventhiran[1,4] , Kiran R Patil[1]

Bidirectional interactions between the immune system and the gut microbiota are key contributors to various physiological functions. Immune-associated diseases such as cancer and autoimmunity, and efficacy of immunomodulatory therapies, have been linked to microbiome variation. Although COVID-19 infection has been shown to cause microbial dysbiosis, it remains understudied whether the inflammatory response associated with vaccination also impacts the microbiota. Here, we investigate the temporal impact of COVID-19 vaccination on the gut microbiome in healthy and immuno-compromised individuals; the latter included patients with primary immunodeficiency and cancer patients on immunomodulating therapies. We find that the gut microbiome remained remarkably stable post-vaccination irrespective of diverse immune status, vaccine response, and microbial composition spanned by the cohort. The stability is evident at all evaluated levels including diversity, phylum, species, and functional capacity. Our results indicate the resilience of the gut microbiome to host immune changes triggered by COVID-19 vaccination and suggest minimal, if any, impact on microbiome-mediated processes. These findings encourage vaccine acceptance, particularly when contrasted with the significant microbiome shifts observed during COVID-19 infection.

## Introduction

In the first 30 mo of the pandemic, there have been reported to be almost 800 million PCR confirmed cases of COVID-19 infection and approaching 7 million related deaths globally (1). To reduce this severity, vaccines were deployed with the aim of promoting anti–SARS-CoV-2 immunity, with almost 13.5 billion vaccine doses administered globally, 150 million of which in the United Kingdom (1). Yet, continued COVID-19 transmission remains of concern (2) with one of the reasons being vaccine hesitancy (3). Thus, data helping to understand holistic effects of vaccination will have a profound impact on the public health management.

The SARS-CoV-2 mRNA and viral vector vaccines induce a strong immune response through the promotion of both innate and adaptive immunities against the spike protein (4). Of note, there has been reported to be a promotion of inflammatory cytokines IFNγ, IL-15, and IL-6 secretion in response to vaccination (5) each known to impact intestinal epithelial architecture and mucosal immunity (6, 7, 8). The sequestration of the spike mRNA by antigen-presenting cells aims to limit the spread into systemic circulation; however, the SARS-CoV-2 spike protein was also found in blood plasma after vaccination (9), potentially leading to inflammation at different sites of the body other than the site of vaccination, including that of the gut. This led us to hypothesize that the systemic immune response to the SARS-CoV-2 vaccination may impact the gut microbiota.

Numerous studies have analysed the gut microbiome during COVID-19 infection (10), finding notable depletion in both commensal bacterial, such as *Bacteroides* and *Bifidobacterium spp.* and beneficial *Lachnospiraceae*, coupled with increased abundances of opportunistic pathogens such as *Streptococcus* and *Clostridium hathawayi*; this indicates a marked dybiosis induced by COVID-19 infection. Yet since the initiation of the vaccination programme against COVID-19, very few studies have addressed the impact of the

[1]Medical Research Council Toxicology Unit, University of Cambridge, Cambridge, UK  [2]Cambridge Institute of Therapeutic Immunology and Infectious Disease, University of Cambridge, Cambridge, UK  [3]Department of Medicine, University of Cambridge, Cambridge, UK  [4]Department of Clinical Immunology, Cambridge University NHS Hospitals Foundation Trust, Cambridge, UK  [5]NIHR Cambridge Clinical Research Facility, Cambridge, UK  [6]NIHR BioResource, Cambridge University Hospitals NHS Foundation Trust, Cambridge, UK  [7]Department of Clinical Immunology, Barts Health, London, UK  [8]UCL GOSH Institute of Child Health Division of Infection and Immunity, Section of Cellular and Molecular Immunology, London, UK  [9]NHS Blood and Transplant, Cambridge, UK  [10]European Molecular Biology Laboratory, Heidelberg, Germany

Correspondence: jedt2@mrc-tox.cam.ac.uk; kp533@mrc-tox.cam.ac.uk
*CITIID-NIHR COVID-19 BioResource Collaboration members are listed in the below Appendix.

**Table 1. Characteristics of the participants in this study.**

| Cohort | Participants | Age | Vaccine | Condition | Treatment |
|---|---|---|---|---|---|
| Healthy controls | 6F 9M | 28–59 Mean = 43.7 | Vaccine doses, n = 20 90% Pfizer doses 10% Moderna doses | NA | NA |
| Immune checkpoint treated cancer patients (ICP) | 9F 26M | 39–86 Mean = 61.7 | Vaccine doses, n = 70 97% Pfizer doses 3% Moderna doses | 11 Metastatic Melanoma 10 Adjuvant Melanoma 5 Melanoma controls 6 Metastatic Renal 3 Renal controls | 3 Nivolumab, 13 Pembrolizumab, 10 Ipilimumab + Nivolumab, 1 Ipilimumab + Pembrolizumab |
| Primary immunodeficient patients (PID) | 4F 5M | 19–61 Mean = 41.1 | Vaccine doses, n = 19 95% Pfizer doses 5% AstraZeneca doses | 1 CD40L deficiency 2 CTLA4 deficiency 4 NFKB1 deficiency 2 Undiagnosed condition | 5 intravenous immunoglobulin 3 Antibiotics |

Participants enrolled in the study are split into one of three cohorts: healthy controls, immune checkpoint treated cancer patients and primary immunodeficient patients. F, female; M, male.

vaccination on the gut microbiome ([11], [12], [13], [14]). Previous work has addressed the link between the gut microbiome and vaccine immunogenicity in which baseline abundances of certain bacterial species before the first vaccine dose have been correlated with a defined end point of vaccine efficacy, typically a vaccine-related readout, such as virus neutralisation or spike-specific antibody titres.

Yet, two open questions remain: how the gut microbiome is affected by COVID-19 vaccination in the days after vaccination when the inflammatory response is at its peak, and are any immediate changes in the gut microbiome maintained or resolved once humoral immunity has been initiated? Addressing this knowledge gap could help understand the extent and the nature of reciprocal links between the gut microbiome and systemic immunity in the context of vaccination. We therefore sought to analyse the gut microbiome of patients receiving doses of the COVID-19 vaccines to decipher whether there were any notable, characteristic changes in the gut microbiome in either healthy or immunocompromised individuals (Table 1). The immunocompromised patients we recruited were cancer patients receiving immune checkpoint-modulating therapies and patients with primary immunodeficiencies. These cohorts consist of patients with impaired immunity resulting from either therapeutic intervention or monogenic defects in immunoregulatory genes (*CTLA4*, *NFKB1*, *CD40L*), respectively.

This presented us with the unique opportunity to elucidate whether the COVID-19 vaccines alter the gut microbiome in the absence of complete, functional immunity and subsequent impairments of the regulation of the gut microbiome.

## Results

### The composition of gut microbiome is not altered by vaccination against COVID-19

To investigate the impact of the COVID-19 vaccines on the gut microbiome, shotgun metagenomic sequencing was performed on a total of 239 fecal samples from 59 patients from our three cohorts (43 healthy control, 160 cancer, and 36 primary immunodeficient

patient samples) (Fig S1). Samples were taken over the course of three vaccine doses, at one of three vaccine timepoints: pre-dose (before vaccination), acute (2–3 d after vaccination) or late (16–28 d after vaccination) for each vaccine dose (Fig 1A). Pre-dose sampling provides a baseline assessment of the gut microbiome before vaccination, whereas acute samples provide the opportunity to assess the effect of vaccination on the gut microbiome at the height of the initial inflammatory response, and late samples allow a determination of any resolution or maintenance of acute effects.

When first observing the $\alpha$-diversities of all samples taken from each of our cohorts (Fig S2A), however when assessing samples taken at different vaccine timepoints from within each cohort, we did not observe any significantly differences (Fig 1B). In agreement with this, when assessing samples from the same patient at different vaccine timepoints using a paired sample analysis (tracking individual patient samples across multiple timepoints, e.g., pre-dose and acute), we also did not see any significant differences in the $\alpha$-diversities of our patient samples (Fig S2B and Table S1). This indicates that the COVID-19 vaccine is not affecting the diversity of the gut microbiome, despite the distinct microbial diversity between the cohorts and individuals.

We next applied principal component analysis to visualize the $\beta$-diversity of our microbiome composition data using the abundance of all detected operational taxonomic units (Fig 1C). The principal components (PCs) describe the largest variation components in the dataset, representing shifts in microbiome composition and potentially reflecting to the abundance changes of bacterial species between the samples. The first five principal components were responsible for the 2.9%, 2.4%, 2.1%, 2.0%, and 1.9% of variation in the data, respectively, and were further analysed using mixed-effect linear models with multiple input variables from our available metadata. In our linear models, we asked whether the vaccination timepoint of the samples in each cohort could improve the explained variance of the PCs when compared with a baseline model describing the explained variance using patient samples as the grouping variable. We found that there was no significant improvement on the baseline model (Fig S2C). Furthermore, a

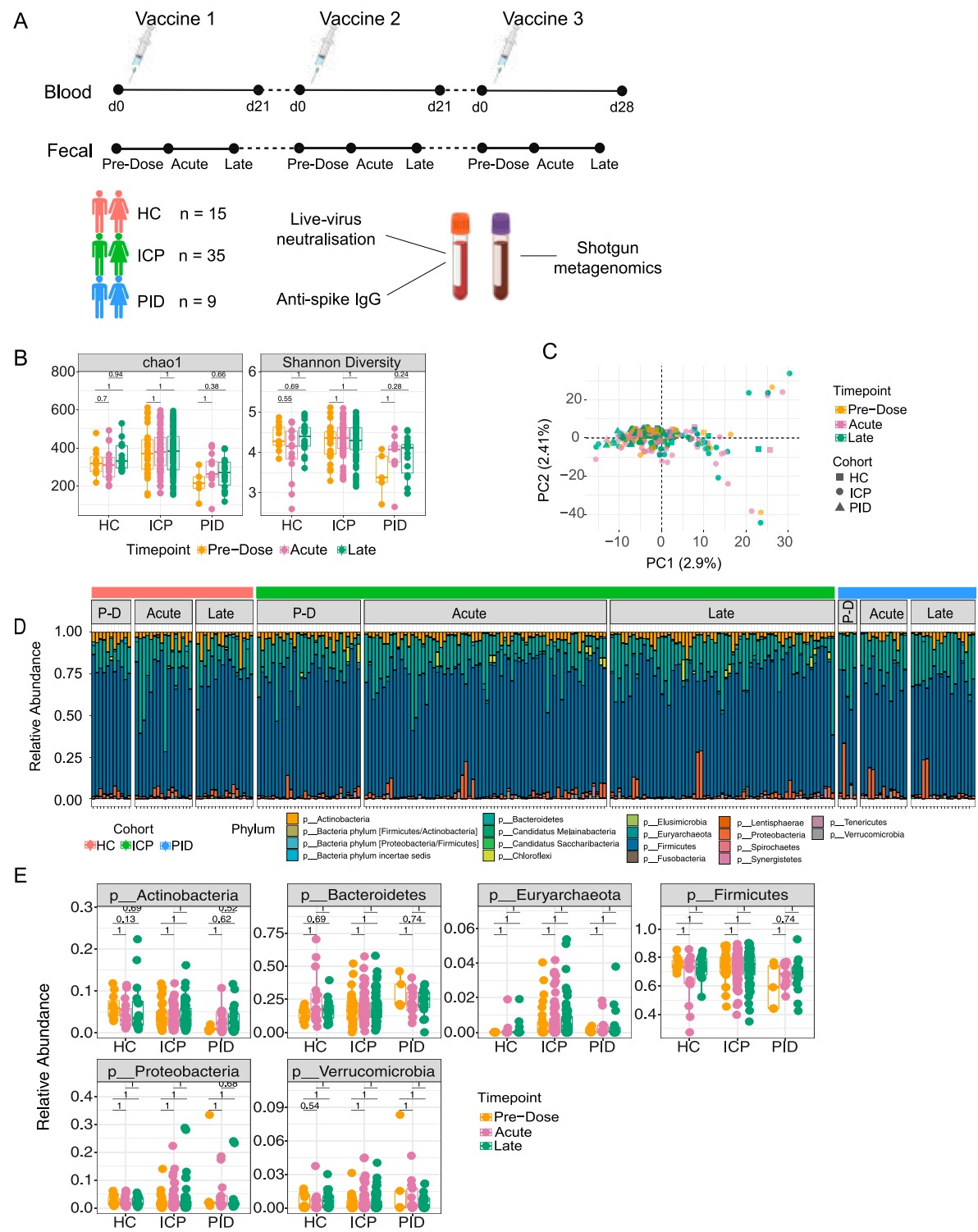

**Figure 1. The composition of the gut microbiome remains unaltered after COVID-19 vaccination.**
**(A)** 59 patients were recruited for longitudinal analysis of the effect of the vaccines against COVID-19. Samples were assigned to one of three cohorts, healthy control, immune-checkpoint therapy treated cancer patients (ICP), or patients with primary immunodeficiencies (PID). Blood samples were analysed for their live-virus neutralisation capacity and quantifying the amount of anti-spike IgG antibodies, whilst fecal samples were analysed with shotgun metagenomics for taxonomic and functional annotations. **(B)** Diversity measures of chao1 and Shannon assessed in fecal samples taken from different vaccine timepoints, from within healthy control, ICP and patients with PID. Statistical testing performed using Wilcoxon test and adjusted for multiple testing using Bonferroni correction. **(C)** Principal component (PC) analysis at the operational taxonomic unit level. Each dot represents a unique sample from within each cohort (shapes) taken at unique timepoints after vaccination (colours). **(D)** Relative abundance at the phyla taxonomic level depicted by colours of each of the bars, from samples taken from each of the cohorts (HC, ICP, and PID),

PERMANOVA analysis did not find the covariate of the timepoints from which the samples were taken to be significantly affecting the microbial composition (Table S2). These analyses together support that the variance we see in our samples is not a signature of the COVID-19 vaccines, rather of the patients presenting with different microbiome compositions themselves.

Next, we asked whether the COVID-19 vaccines induce any changes in the phylum-level composition of the gut microbiome and profiled the relative abundance of phyla across all samples (Fig 1D), observing variation in our patient samples. Moreover, when comparing the top six most prevalent phyla, no significant differences were observed between vaccine timepoints (Fig 1E) despite significant differences in these phyla between cohorts and individuals (Fig S2D). When using a paired analysis approach, we observed only marginal, small effect size, differences (after multiple testing correction) in seven (out of 231) timepoint comparisons in the cancer cohort (Fig S2E) (Table S3). Of those seven, however, only one had a $log_2FC$ greater than 1 in either direction (+1 or −1) (Fig S2F), indicative of notable change in median relative abundance of Lentisphaerae.

This demonstrates that the COVID-19 vaccines do not appreciably alter the composition of the gut microbiome, irrespective of the unique compositions found in our cohort samples.

### COVID-19 vaccination does not induce species level changes in the gut microbiome

We next sought to analyse differentially abundant microbial species between vaccine timepoints, that is, pre-dose, acute, and late, using DESeq2 (15). All cohorts were analysed independently for the abundance changes in samples taken at each timepoint with the most differentially abundant species in a representative heatmap. For the cancer cohort, when assessing the abundance of these top differential-responding bacterial species between samples taken pre-dose and acutely, unsupervised clustering does not demonstrate evident grouping of timepoints (Fig 2A). Among all the species, only two were significantly increased in acute samples compared with the pre-dose samples, *Klebsiella pneumoniae* and *Butyrivibrio crossotus* found in 15 ($P = 1.01 \times 10^{-24}$) and seven samples ($P = 7.63 \times 10^{-12}$) out of the 97 cancer patient samples, respectively (Fig 2B). The former is only representative in a quarter of the cohort (11 patients), only melanoma patients, and within those has an average relative abundance of 0.7% (Fig 2C); the latter in two renal cancer patients, representing on average 3% of the relative abundance (Fig 2D). Considering ~2,500 species are represented across all patient samples, change in two low-abundant and sparsely represented species signifies negligible changes. Similar findings were seen for our other two cohorts, healthy controls and primary immunodeficient patients (Fig S3A–D, respectively). When performing paired sample differential abundance analysis in all three cohorts using DESeq2, we find that no significantly altered species between pre-dose and acute, or pre-dose

and late sample in the healthy control and cancer cohorts. Samples from only one primary immunodeficient patient showed a significant reduction in *Enterobacter sp.* in a late sample compared with pre-dose (Fig S3E). Thus, our findings demonstrate that on a species level there is no unified, biologically relevant change in abundance of microbial species induced by the COVID-19 vaccines.

As the differential abundance analysis considers the change in abundance of all species irrespective of their relative abundance within each sample, we were curious whether there were any noticeable changes in the most abundant species found within each patient cohort that could be attributed to the vaccine timepoints. There was no significant difference in any of the most abundant 15 species in samples taken at any of the three vaccine timepoints (Fig 2E), representing, on average, 47% of the relative abundance of the species within patient samples in the cancer cohort, 50% within the healthy controls, and 53% within the primary immunodeficient patient cohort. This indicates that we see no effect of the vaccine on the species occupying the highest proportion of the microbial niche.

There was considerable concern both at the time of the initial vaccine programme, and to this day, on the safety of COVID-19 vaccines. Given that, we sought to observe if there was any differential outgrowth of bacterial species that have been associated to various immune-related diseases, including gastric cancer and autoimmunity, and metabolic and neurological diseases. Although our study cannot address long-term outgrowth of bacterial associated to these diseases, we are able to highlight if there is any temporal, acute changes in these species which is still of physiological relevance. *Faecalibacterium prausnitzii*, which is reported to be reduced in both gastric cancers, autoimmunity, and Crohn's disease (16), showed no significant abundance changes at the vaccine timepoints in our cohorts (Fig 2F). We also found no presence of *Helicobacter pylori*, which is strongly associated with the initiation and development of gastric cancers (17). *Akkermansia muciniphila*, found to be increased in obesity (18) and correlated with response rates to immune checkpoint blockade therapies in various malignancies (19), was not altered by vaccination in all three cohorts (Fig 2G). In studies of Alzheimer's, *Escherichia coli* has been demonstrated to promote neurodegeneration (20); in our samples we did not see significant difference induced by the vaccine in any of our cohorts (Fig 2H). This supports that the COVID-19 vaccine does not promote the change in abundance of microbes that are associated with various immune-related diseases within any of the three cohorts and is indicative of no greater risk of the aforementioned diseases as a result of COVID-19 vaccination.

### Gut microbiome diversity is not correlated with the magnitude of the response to the COVID-19 vaccines

Within the current literature, a few studies have reported changes in the gut microbiome that correlate with vaccine efficacy (12, 13, 14),

---

separated by the vaccine timepoints from which the sample was taken; PD, Pre-Dose, Acute, and Late. **(E)** Relative abundance of the six most prevalent phyla in patient samples from within each of the cohorts and separated by the vaccine timepoint from which the sample was taken. Statistical testing performed using Wilcoxon test and adjusted for multiple testings using Bonferroni correction. (N = 43 HC, 160 ICP, and 36 PID).

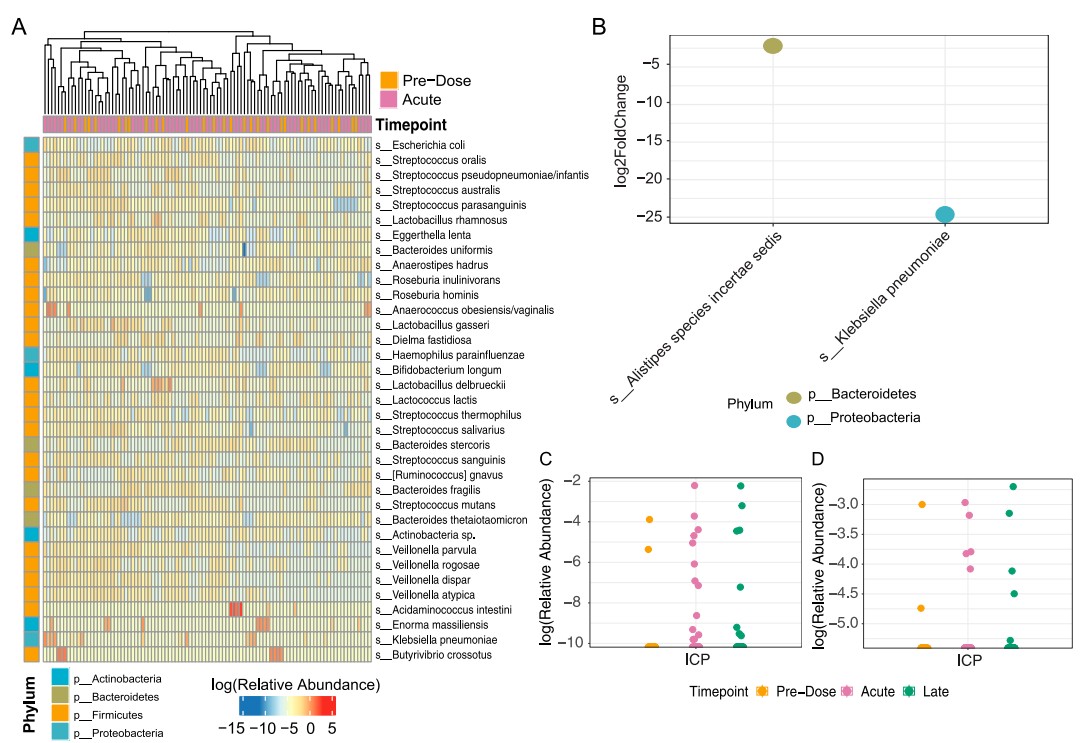

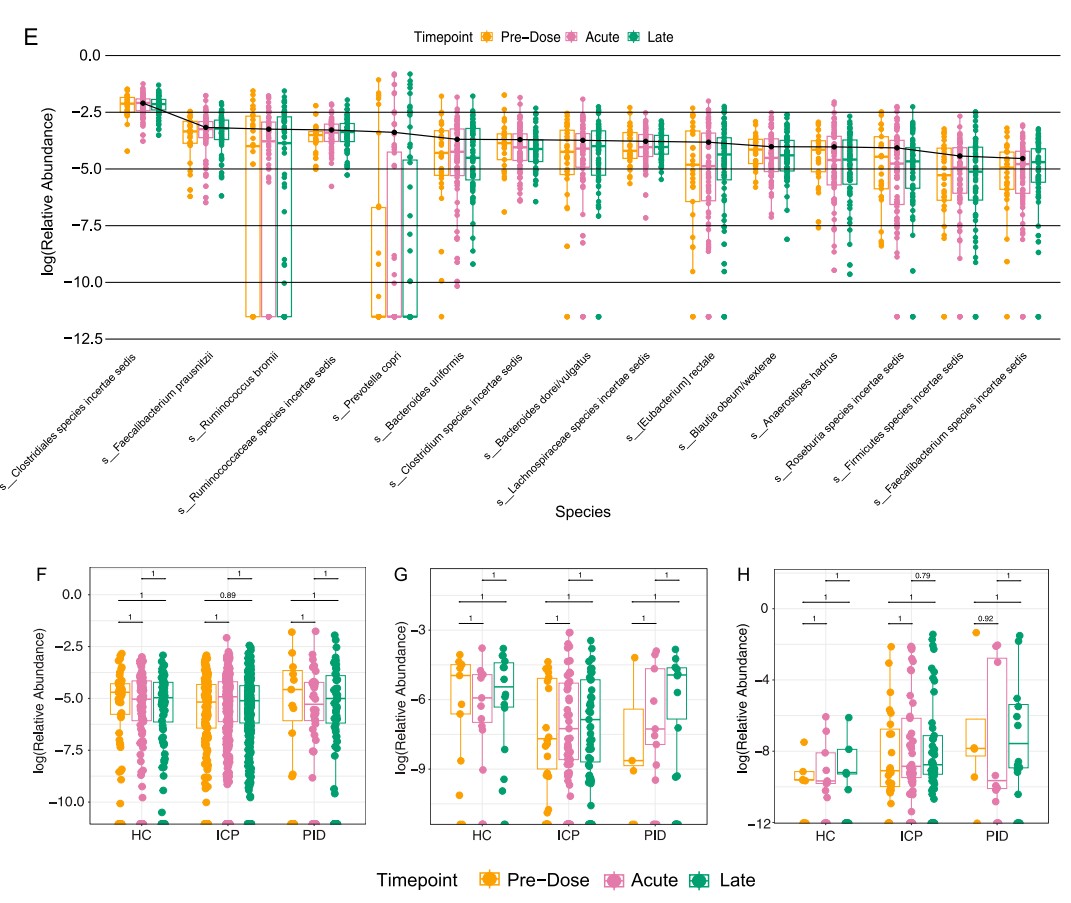

so we sought to determine whether the gut microbiome composition was related to the magnitude of the COVID-19 vaccine response. We performed an assessment of vaccine efficacy using a live-virus neutralisation assay, as a predictive measure of vaccine protection (21), to assess whether vaccine efficacy had correlation with microbial diversity.

When taking the neutralising capacity of patient serum at both the second dose (Fig 3A) and third dose (Fig 3B), we asked whether the Shannon diversity of gut microbiome at different vaccine timepoints was affected by or correlated with neutralisation. We did not see any correlation between diversity and vaccine efficacy in any of our patient cohorts; the same is true for the quantity of anti-spike IgG antibodies (Fig S4). This indicates that the gut microbiome diversity was not correlated with the magnitude of the immune response in our patient cohorts, thus suggesting that improved efficacy of the vaccine does not come at a cost of microbial disturbance.

### The gut microbiome functional capacity was not affected by COVID-19 vaccines

Having investigated the composition and relative abundance of the microbial species that constitute the gut microbiome, we next sought to investigate whether the functional capacity of the microbial species was altered by the COVID-19 vaccines. Using the EggNOG database (22), we assigned functional annotations to the sequenced metagenomes. The highest level of functional annotation depicts three functional groups, cellular processes, and signalling, information storage and processing, and metabolism. In these, we did not see any significant differences between the vaccine timepoints within our cohorts (Fig 4A); similar to taxonomic data presented earlier, when assessing cohort samples separately, there are significant changes (Fig S5A).

We next observed the abundance of the 22 defined functional groups in the next functional annotation level down in the separate vaccine timepoints within each patient cohort (Fig 4B). Representative graphs of the most abundant functional annotations within each of the highest three level functional levels remained unchanged after vaccination within our patient cohorts (Fig 4C), the same is true for the remaining 19 (Fig S5B).

At the lowest functional annotation level, we interrogated the abundance of cluster of orthologous genes at separate vaccine timepoints within our cohorts. Remarkably, no cluster of orthologous genes out of a possible 2,142 genes presented in our patient samples, was significantly altered (when corrected for multiple testing) as a result of the COVID-19 vaccines in any of the three cohorts. This demonstrates that the functional annotations of the gut microbiome are not altered by the administration of the COVID-19 vaccines.

## Discussion

To the best of our knowledge, this study is the first to assess the gut microbiome composition in response to the COVID-19 vaccines across multiple doses and at multiple timepoints with samples taken pre-dose, acutely, and late after vaccination. The study is also the first to assess the effect of the COVID-19 vaccines on the gut microbiome in cancer patients and in patients with inborn errors of immunity associated with severe immune dysregulation. As sampling across the cohorts varied throughout, we opted to combine the three vaccine doses and assess vaccine timepoints or samples from within each cohort. This allowed us to better observe the influence of the COVID-19 vaccines in these contexts.

The relative abundance of microbes within the gut microbiome has more recently been assessed with vaccine immunogenicity including that of vaccines against SARS-CoV-2 virus. The first reported study of the gut microbiome in COVID-19 vaccinated patients correlated vaccine immunogenicity of the inactivated virus, CoronaVac, and the mRNA vaccine-encoding spike protein, BNT162b2 vaccine, with the baseline abundance of gut *Bifidobacterium adolescentis* and *Roseburia faecis*, respectively (12), and noting shifts in microbiome composition. We did not observe changes in these bacterial species, nor the composition, as our study design was to look at changes over vaccine doses individually rather all together. A previous study investigated the variability of the gut microbiome response to the COVID-19 vaccine by correlating RNAseq data with microbial abundance using 16S rRNA gene amplicon sequencing (14). They identified several differentially abundant taxa between high- and low-antibody responders and high- and low-T-cell responders. In the context of immunocompromised cohorts, a previous study assessed patients with inflammatory bowel disease (13) well known to be characterised by gut microbiome dysbiosis (23), who were receiving anti-TNF immunomodulators. Their study did not demonstrate changes in diversity in the above geometric mean vaccine responders but found *Bilophila* abundance correlated to an improved response. What these studies have in common is associating microbiome composition to vaccine immunogenicity; however, these studies analysed single post-vaccination timepoint with potential for confounding effects of various lifestyles and other factors known to profoundly affect microbiome composition such as diet, medications, sampling time of the day, etc. (24, 25, 26, 27). Our multi-timepoint study around individual vaccine dose allowed us to more robustly assess the microbiome changes (or the lack thereof). As the patients enrolled in our study were not all healthy, our cohort samples span a much wider and higher diversity compared with the other study, and all our samples were taken in UK, it remains a

---

**Figure 2. Bacterial species demonstrate minimal change attributable to the COVID-19 vaccines.**
**(A)** Differential abundance analysis using DESeq2 of relative abundance of the top 35 differential species between samples taken at pre-dose (N = 29) and acutely (N = 69) after vaccination. **(B)** Log$_2$ fold-change of the significant differential abundant species taken from the DESeq2 analysis. **(C)** Relative abundance of *Klebsiella pneumoniae* in ICP cohort samples. **(D)** Relative abundance of *Butyrivibrio crossotus* in ICP cohort samples. **(E)** Relative abundance of the top 15 abundant species within the ICP cohort taken at each of the vaccine timepoints (N = 43 HC, 160 ICP, and 36 PID). **(F, G, H)** Relative abundance of various bacterial species correlated with immune-related diseases: *Faecalibacterium prausnitzii* (F), *Akkermansia muciniphila* (G), and *Escherichia coli* (H) within patient samples taken at each vaccine timepoint. Statistical testing performed using Wilcoxon test and adjusted for multiple testing using Bonferroni correction. (N = 43 HC, 160 ICP, and 36 PID).

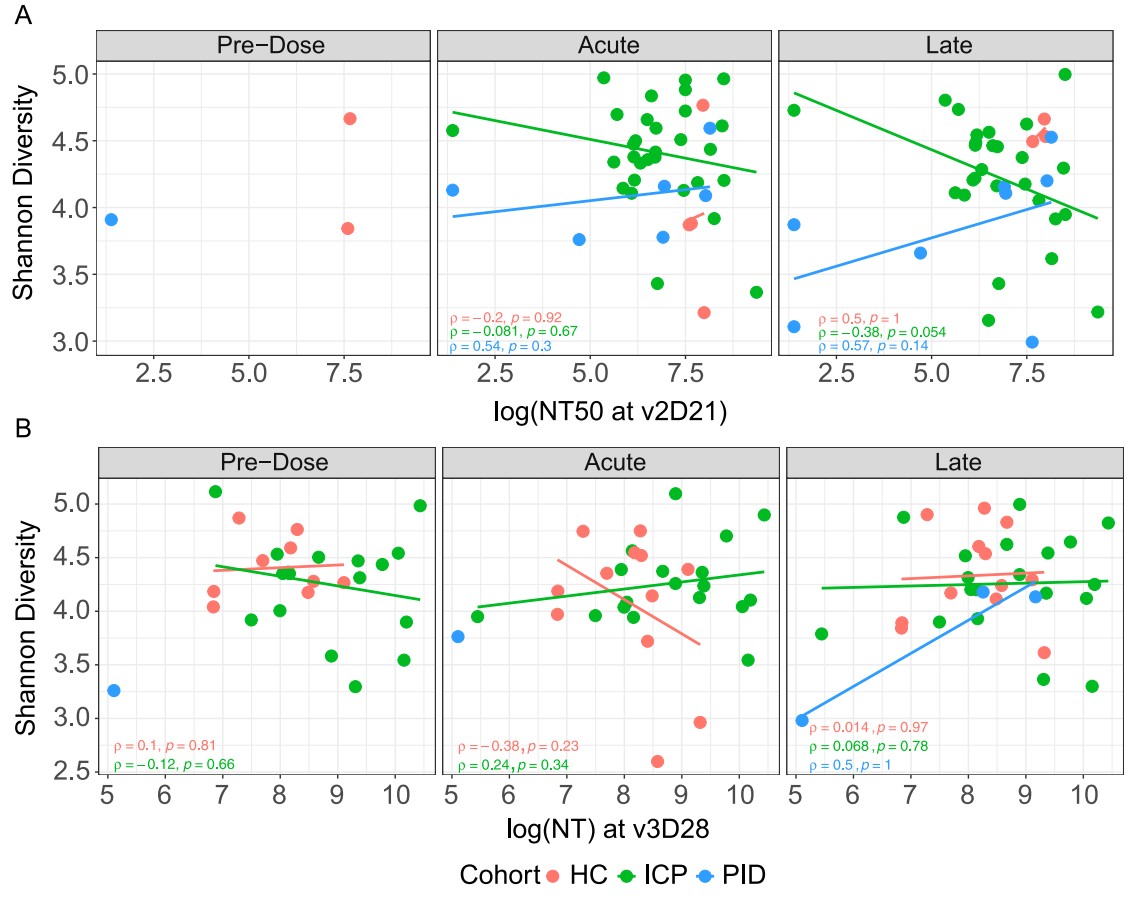

**Figure 3. Vaccine efficacy is not correlated with the gut microbiome diversity.**
Live-virus neutralisation capacity (NT) assessed against Shannon diversity of fecal samples, each point represents a different sample taken at one of the three vaccine timepoints. Colours represent cohorts, within healthy control, immune-checkpoint therapy-treated cancer patients (ICP) and patients with primary immunodeficiencies (PID). Correlated vaccine response through neutralisation capacity of patient serum taken at the peak of the second dose (v2D21) (N = 9 HC, 57 ICP, and 15 PID). **(A)** or third dose (v3D28) (N = 33 HC, 54 ICP, and 5 PID). **(B)** rho and *P*-values from Spearman's Rank correlation testing displayed.

possibility that country-specific differences explain the variance in the findings across the studies.

In our patient cohorts, we did not find any significant effect on the diversity of the gut microbiome after COVID-19 vaccination despite considerable differences between the cohorts. In line with the studies investigating the gut microbiome of patients with primary immunodeficiencies (28), we also observed decreased diversity in our cohort compared with control samples. These patients have been demonstrated to exhibit increased gut permeability with higher rates of bacterial translocation (29), perhaps indicative that there is bidirectional permeability of systemic immunity affecting the gut microbiome. In our study, we find no influence on microbiome variation after vaccination in patients at a genetically determined persistent state of immune dysregulation.

In melanoma patients, the presence of bacterial species from the *Actinobacteria* and *Firmicutes* phyla have been associated with better responses to immune checkpoint blockade therapies (30, 31, 32). An *A. muciniphilia* signature was also found in renal cancer patients responding better to immune checkpoint blockade therapy (33, 34). Although cancer progression is reported to be linked to

gut microbiome composition and its derived metabolites, these associations vary between cancer types (35). We saw wide compositional variation within our cancer cohort samples, perhaps because of the wide range of disease presentation and treatment included in our patients. Our study did not consider factors such diets and medications known to affect the gut microbiome (36). Nevertheless, this is not critical as our analysis suggests stability as opposed to specific changes and post-hoc power calculation indicates sufficient power against false negatives when taking paired samples (Pre-dose-Acute, Pre-dose-Late and Acute-Late) (for effect size, Cohen's d = 0.55, estimated power = 0.8). Cancer patients' response to vaccination depends significantly on cancer type, for example, antibody-related immune responses in solid cancers are better than in haematological cancers (37). It is therefore notable that in an immunologically diverse cohort of individuals, with varied vaccine responses, we did not observe any effect of the COVID-19 vaccination on the gut microbiome, indicating stability irrespective of pre-dose composition.

Although we did not observe changes at any taxonomic level or functional capacity, we cannot rule out genetic changes at mutational levels that may alter the microbiota function. An

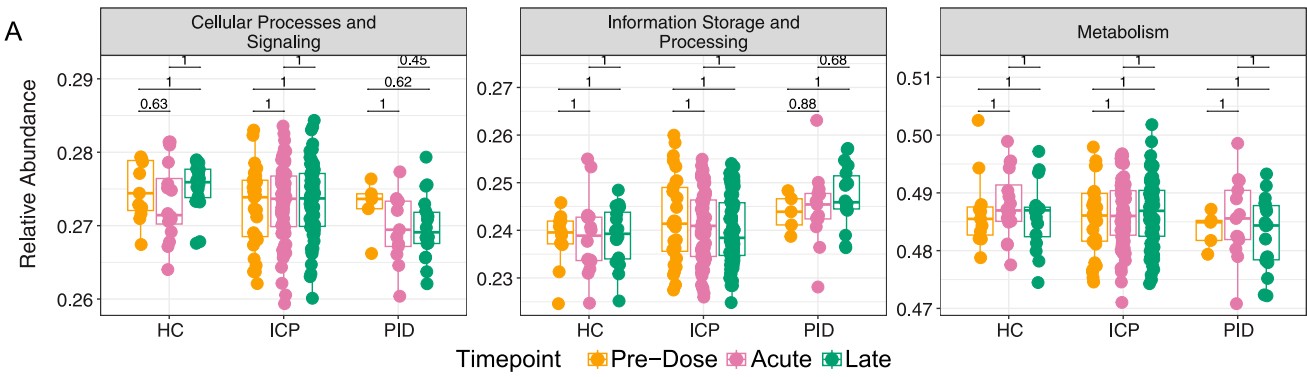

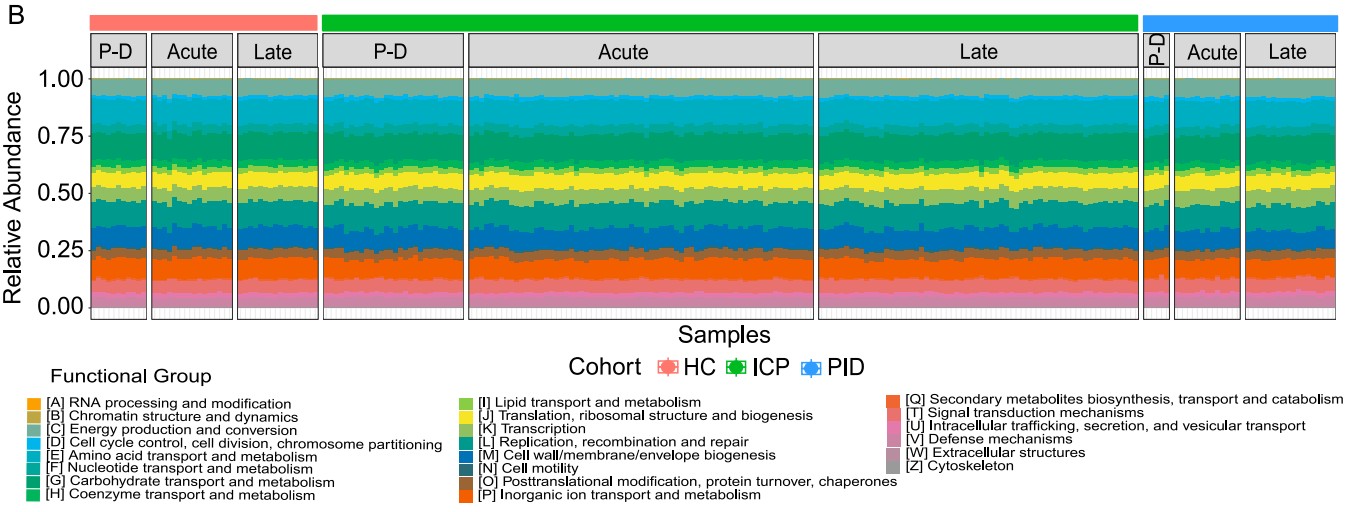

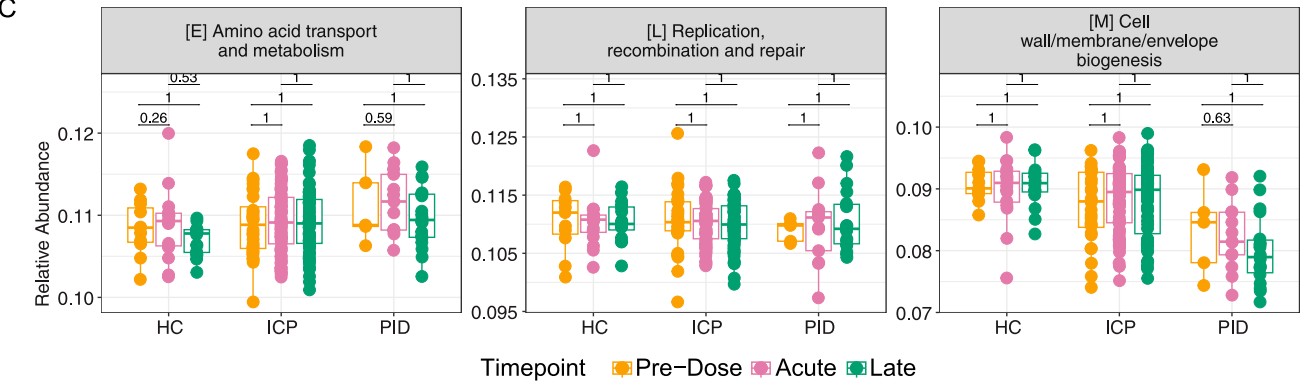

**Figure 4. Functional capacity of microbiome samples are not altered by the COVID-19 vaccines.**
**(A)** The relative abundance of the highest functional annotation level using the EGGNOG database within patient samples at different vaccine timepoints in each of our patient cohorts. **(B)** Functional composition depicted by colours of each of the bars, from samples taken from each of the cohorts (HC, ICP, and PID), separated by the vaccine timepoints from which the sample was taken; PD, Pre-Dose, Acute, and Late. **(C)** Relative abundance of the three most abundant functional annotations in our patient samples from within each of our patient cohorts and separated by the vaccine timepoint from which the sample was taken. Statistical testing performed using Wilcoxon test and adjusted for multiple testing using Bonferroni correction (N = 43 HC, 160 ICP, and 36 PID).

independent functional validation such as metabolomics to look for bacterial derived short-chain fatty acids, tryptophan, and bile acid metabolites known to mediate microbiome–host interactions (38) could be used to assess this.

Sampling from the primary immunodefficient cohort was limited reflecting the rarity of the individuals within the general population. Despite this, we were able to recruit patients characterised by monogenic defects in both intrinsic and extrinsic B-cell aetiologies,

thus representing multiple facets within the rare primary immunodeficient population. Furthermore, we were unable to sample all patients at all vaccine timepoints across all vaccine doses. Nevertheless, our findings still bare relevance as we assess patient cohorts individually and where possible the paired sample data analysis aligns with the overall findings.

Although the global vaccination efforts have controlled the spread of the SARS-CoV-2 virus, there were still reported to be 2.6 million new cases within the past month (1), highlighting that prevention of disease through vaccination is still relevant for public health. Considering the measurable impact of common life factors such as alcohol consumption, meat intake, and commonly used medications on the microbiota (38), our study finds that the vaccination has negligible, if any, impact on microbiome-mediated processes. The contrast is even starker when considering large microbiome changes have been reported for COVID-19 infection (10). Our findings indicate that the gut microbiome remains stable postvaccination and provides an additional reassurance towards promoting vaccine uptake.

# Materials and Methods

### Study recruitment and ethics

Participants volunteered for an observational study and were enrolled to one of 3 cohorts, healthy controls, cancer patients which presented with either melanoma or renal malignancies, or primary immunodeficient patients with defined mutations in key immunoregulatory genes, and patients with clinical presentation aligning to that of defined primary immunodeficiency, including antibody deficiency (Table 1). Almost all participants in each cohort received BNT162b2 Pfizer vaccine, aside from one patient at second dose receiving AstraZeneca, and six patients receiving Moderna at the third (Table 1). Patients were excluded if presenting with positive COVID-19 serology or if presenting in hospital with clinal symptoms/features related to their disease which may influence the physiological response to the COVID-19 vaccination. The research was conducted in accordance with the principles of Good Clinical Practice and following approved protocols of the NIHR National Bioresource. Samples were collected at NIHR Cambridge Clinical Research Facility, Addenbrookes Hospital, Cambridge, UK. Samples were provided with the written informed consent of all study participants under the NIHR National BioResource - Research Tissue Bank (NBR-RTB) Ethics (Research Ethics Committee [REC]:17/EE/0025). All participants were consented under the East of England Cambridge South national consent (REC:13/EE/0325).

### Participant sampling

Participant samples were anonymised by clinical staff before sample delivery to the research laboratory. Peripheral blood and fecal samples were collected longitudinally over the course of up to three doses of the vaccines against COVID-19 (sample coverage varied across doses [Fig S1]). Samples were taken from 9th February, 2021, to 22nd February, 2022. PBMCs were extracted from blood samples using

density gradient centrifugation, stored temporarily at –80°C, before being transferred to long-term storage in liquid nitrogen. Serum was isolated from peripheral blood via centrifugation and stored at –80°C until required. Fecal samples were collected at three timepoints around each vaccine dose: pre-dose (94% of samples taken within 3 d before vaccination, the remaining three samples taken up to 14 d prior), acute (day 2 or 3 after vaccination) or late (day 16–28 after vaccination). Fecal matter was collected in OMNIgene•GUT kits (DNA Genotek) whereby samples are stored in a stabilizing, inactivating solution. Samples were transported to the laboratory and homogenized upon arrival before being stored at –80°C until required.

### Fecal DNA extraction and sequencing

DNA was extracted from fecal samples using QIAamp PowerFecal Pro DNA kits (QIAGEN). Samples were thawed and ~250 mg of fecal sample was lysed via bead beating. According to the kit protocol, the sample was then cleaned of non-DNA organic and inorganic material, and then washed using ethanol. DNA was eluted into 10 mM Tris and quantified using the Qubit 1X dsDNA High-Sensitivity (HS) Assay Kit (Thermo Fisher Scientific) using the Qubit fluorometer (Thermo Fisher Scientific). DNA at a concentration of 10 mg/μl was sent for sequencing. Samples were sent in bulk for sequencing blinded to sample timepoints of each patient. Shotgun metagenomic sequencing was performed with Illumina NextSeq 2000 sequencing platform using paired-end reads of 150 bp in length, resulting in a median of ~28 million reads per sample.

### Shotgun metagenomic analysis

Raw sequencing data were preprocessed with PRINSEQ++ (39 *Preprint*) (v.1.2.4) in paired read mode, quality trimming to a minimal Phred score of 30 in a window of 15 bases and removing reads of less than 75 bp length after trimming. In addition, host contamination was removed by mapping against the GRCh38 reference human genome using Bowtie2 (40) (v.2.4.5) and removing any mapped reads from the dataset.

Raw, trimmed, and filtered reads were checked for quality using FastQC (v.0.11.9) (https://www.bioinformatics.babraham.ac.uk/projects/fastqc/). From the remaining read pairs, taxonomic profiling was determined using mOTUs3 (41) (v.3.0.1) profiler. For functional profiling, the remaining read pairs after filtering were assembled using metaSPAdes (42) (v.3.15.4) with a *k-mer* size of 55. The resulting scaffolds were filtered for at least 200-bp length and weighted by their average coverage of the filtered reads to enable quantitative analysis. The remaining scaffolds were aligned to the EggNOG database (22) (downloaded on 2022/04/08) using DIAMOND (43) (v.2.0.13). Microbiome analysis was performed in R (v.4.2.2) using relative abundances, within the phyloseq (44) (v.1.42.0), microbiome (45) (v.1.20.0) and vegan (46) (v.2.6-4) packages. Differential abundance analysis was performed using DESeq2 (15) (v.1.38.3).

### Serological assessment of immune response to COVID-19 vaccines

Serum samples were thawed, heat-inactivated at 56°C for 30 min. Measurements for the dilution of serum that reduces viral activity by 50% ($NT_{50}$) against WT SARS-CoV-2 were as previously reported

(47, 48). For anti–SARS-CoV-2–specific IgG antibodies, we used a previously described method (49, 50), in which Luminex bead sets are covalently coupled to the recombinant SARS-CoV-2 proteins nucleocapsid protein, spike, and receptor-binding domain to quantify antibody levels.

### Statistical analysis

Wilcoxon tests with multiple testing corrections using the Bonferroni test were deployed throughout, using pairwise comparison where appropriate. Mixed-effect linear modelling was performed using lmer4 (51) in R. Correlation was determined using Spearman's Rank coefficient.

## Appendix: Consortia CITIID-NIHR COVID—19 BioResource Collaboration

Juan Carlos Yam-Puc, Zhaleh Hosseini, Emily C Horner, Nonantzin Beristain-Covarrubias, Robert Hughes, Maria Rust, Rebecca H Boston, Lucy H Booth, Edward Simmons-Rosello, Magda Ali, Lakmini Kahanawita, Anne Elmer, Caroline Saunders, Areti Bermperi, Sherly Jose, Nathalie Kingston, Thomas E Mulroney, Sarah Spencer, Nicholas J Matheson and James ED Thaventhiran.

## Data Availability

Sequencing data is available in the European Genome-phenome Archive (Accession number: EGAD50000000258). The code is available on GitHub: https://github.com/RHBoston/COVID-19_Vaccination_GM.

## Supplementary Information

## Acknowledgements

We are grateful to the clinical staff at NIHR Cambridge Clinical Research Facility and Addenbrookes hospital who facilitated patient recruitment and sample collection, and the members of the Thaventhiran and Patil laboratory for help in processing the samples. This work was funded by the UK Medical Research Council (project numbers MC_UU_00025/11 and MC_UU_0025/12) and Addenbrookes Charitable Trust Grant (ACT 900254).

### Author Contributions

RH Boston: data curation, formal analysis, validation, investigation, visualization, and writing—original draft, review, and editing.
R Guan: data curation, software, formal analysis, visualization, and writing—original draft, review, and editing.
L Kalmar: software and formal analysis.
S Beier: software and formal analysis.
EC Horner: investigation.
N Beristain-Covarrubias: investigation.
JC Yam-Puc: investigation.
P Pereyra Gerber: investigation and methodology.
L Faria: investigation and methodology.
A Kuroshchenkova: investigation and methodology.
AE Lindell: investigation and methodology.
S Blasche: investigation and methodology.
A Correa-Noguera: resources.
A Elmer: resources.
C Saunders: resources.
A Bermperi: resources.
S Jose: resources.
N Kingston: resources.
S Grigoriadou: resources.
E Staples: resources.
MS Buckland: resources.
S Lear: resources.
NJ Matheson: resources.
V Benes: resources and methodology.
C Parkinson: resources.
JED Thaventhiran: conceptualization, resources, funding acquisition, and writing—review and editing.
KR Patil: conceptualization, resources, supervision, funding acquisition, methodology, and writing—original draft, review, and editing.

### Conflict of Interest Statement

The authors declare that they have no conflict of interest.

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
