## [Reviewer comments · Life Science Alliance]

Life Science Alliance

Stability of gut microbiome after COVID-19 vaccination in healthy and immuno-compromised individuals

Rebecca Boston, Rui Guan, Lajos Kalmar, Sina Beir, Emily Horner, Nonantzin Beristain-Covarrubias, Juan Carlos Yam-Puc, Pehuén Pereyra Gerber, Luisa Faria, Anna Kuroshchenkova, Anna Lindell, Sonja Blasche, Andrea Correa-Noguera, Anne Elmer, Caroline Saunders, Areti Bermperi, Sherly Jose, Nathalie Kingston, CITIID-NIHR COVID-19 BioResource Collaboration, Sofia Grigoriadou, Emily Staples, Matthew Buckland, Sara Lear, Nicholas Matheson, Vladimir Benes, Christine Parkinson, James Thaventhiran, and Kiran Patil

DOI: <https://doi.org/10.26508/lsa.202302529>

Corresponding author(s): Kiran Patil, University of Cambridge and James Thaventhiran, University of Cambridge

Review Timeline:	Submission Date:	2023-12-13
	Editorial Decision:	2023-12-19
	Revision Received:	2023-12-27
	Accepted:	2023-12-29

Transaction Report:

Please note that the manuscript was previously reviewed at another journal and the reports were taken into account in the decision-making process at *Life Science Alliance*.

December 19, 2023

RE: Life Science Alliance Manuscript #LSA-2023-02529-T

Dr Kiran Raosaheb Patil
University of Cambridge
MRC Toxicology Unit
Tennis court road
Cambridge, Cambridgeshire CB21QR
United Kingdom

Dear Dr. Patil,

Thank you for submitting your revised manuscript entitled "Gut microbiome remains stable following COVID-19 vaccination in healthy and immuno-compromised individuals". We would be happy to publish your paper in Life Science Alliance pending final revisions necessary to meet our formatting guidelines.

- please upload your main manuscript text as an editable doc file
- please upload your main and supplementary figures as single files
- please add ORCID ID for the corresponding (and secondary corresponding) author--you should have received instructions on how to do so
- please add a Summary Blurb/Alternate Abstract to our system
- please add Keywords and a Category for your manuscript to our system
- please add the Twitter handle of your host institute/organization as well as your own or/and one of the authors in our system
- please consult our manuscript preparation guidelines <https://www.life-science-alliance.org/manuscript-prep> and make sure your manuscript sections are in the correct order
- please add authors' contributions to the system as well
- Tables should be uploaded in editable .doc or excel format and included at the bottom of the main manuscript file or sent as separate files.
- please add your main, supplementary figure, and table legends to the main manuscript text after the references section
- please update the Data Availability statement to include accession information for the sequencing data. The GitHub link also needs to be updated.

A. FINAL FILES:

B. MANUSCRIPT ORGANIZATION AND FORMATTING:

Sincerely,

December 29, 2023

RE: Life Science Alliance Manuscript #LSA-2023-02529-TR

Kiran Patil
University of Cambridge
United Kingdom

Dear Dr. Patil,

Thank you for submitting your Research Article entitled "Stability of gut microbiome after COVID-19 vaccination in healthy and immuno-compromised individuals". It is a pleasure to let you know that your manuscript is now accepted for publication in Life Science Alliance. Congratulations on this interesting work.

DISTRIBUTION OF MATERIALS:

Again, congratulations on a very nice paper. I hope you found the review process to be constructive and are pleased with how the manuscript was handled editorially. We look forward to future exciting submissions from your lab.

Sincerely,
